# EXPLAINABLE MIXTURE MODELS THROUGH DIFFERENTIABLE RULE LEARNING

**Matthias Wilms**[*]**, Sascha Xu**[*]**, Jilles Vreeken**
CISPA Helmholtz Center for Information Security,
Saarbrücken, Germany
{mwilms,sascha.xu,jv}@cispa.de

## ABSTRACT

Mixture models excel at decomposing complex, multi-modal distributions into simpler probabilistic components, but provide no insight into the conditions under which these components arise. We introduce explainable mixture models (XMM), a framework that pairs each mixture component with a human-interpretable rule over descriptive features. This enables mixtures that are not only statistically expressive but also transparently grounded in the underlying data. We formalize the problem and examine conditions under which an XMM exactly captures a target distribution. We then propose a scalable, differentiable learning procedure for discovering sets of rules. Experiments on synthetic and real-world datasets demonstrate that our method discovers interesting sub-populations in both univariate and multivariate settings, offering interpretable insights into the structure of complex distributions.

## 1 INTRODUCTION

Mixture models represent complex, multi-modal data as combinations of simpler distributions (McLachlan et al., 2019). On a dataset of insurance charges (Choi, 2017) for example, a Gaussian mixture model (GMM) identifies subpopulations with distinct modes as shown in Fig. 1. In many applications, however, we also have access to descriptive features, e.g. age or BMI. Classical mixture models fit the marginal distribution of the target and therefore cannot leverage such features to explain when different sub-distributions arise.

To overcome this limitation, conditional density estimation (CDE) extends mixtures by modeling the conditional distribution of outcomes given features. In particular, mixture density networks (Bishop, 1994) and kernel mixture networks (Ambrogioni et al., 2017) parameterize mixture weights and components as functions of descriptive features. However, these dependencies are typically modeled with neural networks that do not yield human-interpretable explanations. More broadly, CDE methods tend to prioritize predictive accuracy over interpretability (Sugiyama et al., 2010). While tree-based approaches (Cousins & Riondato, 2019; Yang & van Leeuwen, 2024) offer insight, we observe in experiments they can overfit and lack support for overlapping regions.

---

[*]Equal contribution.

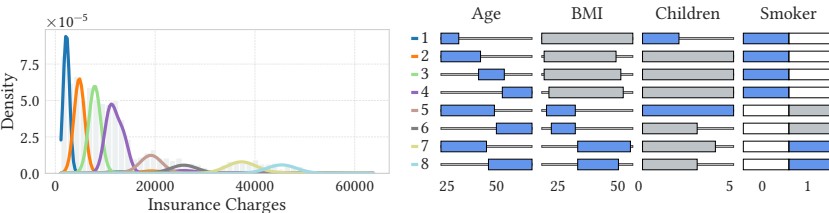

Figure 1: The XMM recovers similar modes to the GMM (left), and also explains when each mode is observed using simple rules over descriptive features.

To this end, we propose Explainable Mixture Models (XMM), a framework that directly pairs each mixture component with a human-interpretable rule over descriptive features. The XMM framework defines each mixture component as a data-induced distribution rather than restricting it to a particular parametric family, e.g., Gaussian, and naturally allows for overlapping components. On the insurance dataset, a fitted XMM in Fig. 1 recovers near identical modes to the GMM (see Appx. A), but add the same time also provides simple, interval-based rules that explain when each mode is observed. The subpopulation with the lowest insurance charges, for example, corresponds to young, non-smoking individuals without children, whereas the highest-charge component comprises older, smoking individuals with high BMI. Our main contributions are as follows:

1. **Concept.** We propose XMM, which both characterize the subpopulations of the global distribution, whilst accurately estimating the local conditional density given any feature vector.

2. **Theory.** We derive exact-recovery conditions for marginal and conditional densities and introduce regularizers to steer learning towards these regimes.

3. **Practice.** We propose a scalable, differentiable training procedure and show that the XMM accurately models the underlying distribution whilst discovering interesting subpopulations.

## 2 RELATED WORK

Mixture models are a classical tool for density estimation and clustering. There exist many variants based on parametric families such as Gaussians (Reynolds, 2015) or t-distributions (Peel & McLachlan, 2000) as well as nonparametric approaches (Antoniak, 1974). In general, unconditional mixture models however are limited to modeling latent component variables (Viroli & McLachlan, 2019).

**Mixture of Experts** (MoE) (Jacobs et al., 1991) are a general class of models in which a gating network determines the weighting of local experts. While MoEs typically rely on black-box neural networks for gating, recent surveys identify interpretability as a critical open challenge (Mu & Lin, 2025). Interpretable variants have been proposed (Ismail et al., 2023; Pradier et al., 2021), however, these approaches focus primarily on classification or deferral to human experts. EMMs share the high-level conditional mixture structure of MoEs but differ fundamentally by targeting conditional density estimation through differentiable rule learning. Similarly, Conditional VAEs (CVAE) (Sohn et al., 2015) can model complex conditional distributions $p(y|x)$, However, they rely on a latent prior $z$ and deep neural networks, resulting in a black-box model. In contrast, EMMs explicitly model the conditional density through rule-based components, providing direct insight into structure of the data without latent variables.

**Conditional density estimation (CDE)** aims to estimate the full conditional distribution of a target variable $y$ given input features $x$. Approaches range from kernel and RKHS-based estimators (Hyndman et al., 1996; Sugiyama et al., 2010), random forest (Watson et al., 2023), to normalizing flows (Winkler et al., 2019). Another relevant line of work extends mixture models to the conditional setting. In Mixture Density Networks (Bishop, 1994) a neural network outputs mixture parameters as functions of $x$, while Kernel Mixture Networks (Ambrogioni et al., 2017) replace the parametric mixture components with nonparametric kernels. However, both methods use black-box neural networks for gating and thus do not provide insight into *when* each component is active, a limitation shared by all the aforementioned CDE methods.

To address this issue, interpretable, tree-based approaches have been proposed. Density Estimation Trees (Ram & Gray, 2011) use interpretable tree structures but only target the unconditional density. CADET uses trees to model conditional densies with exponential family distributions in the leaves (Cousins & Riondato, 2019), but tends to learn very deep trees that are hard to interpret. Most similar to our approach is CDTREE (Yang & van Leeuwen, 2024), which learns a minimum description length regularized decision tree with a histogram in each leaf. Both approaches however are primarily aimed at fitting densities, and not for discovering its components. Consequently, these approaches can yield trees with many fine-grained leaves, limiting interpretability in practice.

**Subgroup discovery** is a line of work where interpretability is key but with an objective orthogonal to mixture modelling (Atzmueller, 2015). The goal is to identify a subpopulation, e.g. a single mixture component, that is statistically interesting with respect to a target variable and describe it through a human-interpretable rule. Using combinatorial (Lavrač et al., 2004; Atzmueller & Puppe,

2006) or continuous optimization (Xu et al., 2024), a rule is learned that maximizes the measured deviation of the subgroup from the global population (Todorovski et al., 2000).

The main difference to XMM is that subgroup discovery is inherently local, focusing on isolating an interesting subset of the data rather than modeling the entire population. While there exist approaches that learn multiple subgroups (Van Leeuwen & Knobbe, 2012; Proença et al., 2022), they typically do not attempt to model the full conditional distribution. XMM extend the notion of a subgroup to mixture models: each component represents a subgroup of the data characterized by an interpretable rule, while the model collectively captures the entire conditional distribution.

**Summary.** Explainable Mixture Models bring together ideas from all three areas: In contrast to neural gated mixture models, XMM provide interpretable rules for each component; Compared to subgroup discovery, we model the entire domain; And compared to tree-based CDE, we allow for a mixture of components rather than a single tree. In the following, we will formally define Explainable Mixture Models and show how to learn them from data.

## 3 EXPLAINABLE MIXTURE MODELS

We consider a dataset of $n$ pairs $\{(\mathbf{x}^{(l)}, y^{(l)})\}_{l=1}^n$ consisting of a **descriptive feature vector** $\mathbf{x} \in \mathbb{R}^d$ of $d$ real-valued features and a **target value** $y \in \mathcal{Y}$. We assume each sample $(\mathbf{x}, y)$ to be a realization of a pair of random variables $(X, Y) \sim P_{X,Y}$, drawn i.i.d. We write $p$ to denote probability density functions and $P$ to denote probability distributions.

Our goal is to explain the distribution of the target variable $Y$ as a mixture of simpler components. The idea is to use components that are grounded in a human-interpretable explanation over the descriptive features $X$ instead of being latent factors. An explainable mixture model (XMM) therefore not only provides a decomposition of the target into simpler sub-distributions, but also explains the conditions under which these sub-distributions are observed.

**Definition 1 (Marginal-XMM)** *An explainable mixture model $\mathcal{M} = \{e_i\}_{i=1}^k$ of the marginal density $p(y)$ is defined as a set of $k$ feature-based explanations $e_i : \mathbb{R}^d \to \{0, 1\}$ with non-zero support, i.e. $\mathbb{E}[e_i(X)] > 0$. For each respective explanation $e_i$, we define the mixture weight $w_i$ as*

$$w_i = \frac{\mathbb{E}[e_i(X)]}{\sum_{j=1}^k \mathbb{E}[e_j(X)]} \,, \tag{1}$$

*where it holds that $w_i \geq 0$ and $\sum_{i=1}^k w_i = 1$. The induced density $p_{\mathcal{M}}(y)$ is a finite mixture of $k$ components as per*

$$p_{\mathcal{M}}(y) = \sum_{i=1}^k w_i \, p_i(y) \,, \qquad p_i(y) := p_{Y \mid (e_i(X)=1)}(y) \,. \tag{2}$$

The marginal XMM as a weighted sum of simpler component densities $p_i(y)$, based on the standard finite mixture model (McLachlan et al., 2019). The differentiating factor of an XMM compared to conventional mixture models lies in the explainability of the individual components $p_i(y)$. Instead of restricting these to a parametric family, e.g. Gaussians, an XMM is based on non-parametric, *data-induced* densities $p_i(y)$. Each component reflects the conditional distribution of the target $Y$ given that the explanation $e_i$ over the descriptive features $X$ holds. The choice of human-interpretable explanation $e_i$ (e.g., logical rules) is application dependent and agnostic to the definition.

**Proposition 1** *Let $\mathcal{M} = \{e_i\}_i^k$ be an XMM with a marginal density as per Def. 1. If the set of explanations $e_i$ form a partition of the feature space $\mathbb{R}^d$, i.e. $\sum_{i=1}^k e_i(\mathbf{x}) = 1$ for all $\mathbf{x}$ in the support of $P_X$, then the induced density $p_{\mathcal{M}}(y)$ equals the true marginal density $p_Y(y)$.*

Proposition 1 is a direct consequence of the law of total probability (See for the proof Appx. B.1). The result shows that we cannot rely on maximization of the marginal likelihood to learn an XMM. Setting all components to the same constant function, e.g. $e_i(\mathbf{x}) = 1$ for all $i$, leads to a perfect fit of the marginal distribution, provided that the component densities $p_i(y)$ are sufficiently flexible. Therefore, we cannot expect to learn a meaningful XMM by maximizing the marginal likelihood. To address this issue, we next introduce a conditional interpretation of the XMM.

## 3.1 Conditional Explainable Mixture Models

The issue of treating an XMM as a purely marginal model is the degeneracy of maximum likelihood solutions. To address this, we leverage the ability of an XMM to explain under which conditions distinct sub-distributions occur and formally introduce the conditional XMM to model the conditional density $p_{Y|X}(y \mid \mathbf{x})$.

**Definition 2 (Conditional-XMM)** *An explainable mixture model $\mathcal{M} = \{e_i\}_{i=1}^k$ of the conditional density $p(y \mid \mathbf{x})$ is defined as a set of $k$ explanations $e_i : \mathbb{R}^d \to \{0, 1\}$ with non-zero support, i.e. $\mathbb{E}[e_i(X)] > 0$, and complete coverage, i.e. $\sum_{i=1}^k e_i(\mathbf{x}) > 0$ for all $\mathbf{x}$ in the support of $P_X$. For each feature vector $\mathbf{x}$, we define the conditional mixture weights $w_i(\mathbf{x})$ as*

$$w_i(\mathbf{x}) = \frac{e_i(\mathbf{x})}{\sum_{j=1}^k e_j(\mathbf{x})} \ . \tag{3}$$

*The induced conditional density $p_{\mathcal{M}}(y \mid \mathbf{x})$ is a finite mixture of $k$ components, where*

$$p_{\mathcal{M}}(y \mid \mathbf{x}) = \sum_{i=1}^k w_i(\mathbf{x}) \, p_i(y) \ , \qquad p_i(y) \coloneqq p_{Y \mid (e_i(X)=1)}(y) \ . \tag{4}$$

Similar to marginal models, the conditional XMM is a finite mixture of simpler component densities $p_i(y)$. In addition, the mixture weights $w_i(\mathbf{x})$ are now dependent on the descriptive features, similar to a mixtures-of-experts (MoE) model (Jacobs et al., 1991). The main difference to MoEs is that an XMM consists of explanation-based components, which are derived from the data, while in a MoE, any gating mechanism is permissible and with arbitrary parametric experts that need not represent an underlying demographic group. Next, we examine what conditions are needed so that a mixture $\mathcal{M}$ faithfully represents the true conditional distribution.

**Proposition 2** *Let $\mathcal{M} = \{e_i\}_i^k$ be an XMM with a conditional density as per Def. 2. If the set of explanations $e_i$ form a partition of the feature space $\mathbb{R}^d$ into homogeneous regions with respect to the target variable $Y$, i.e. for every explanation $e_i$ and its induced sub-distribution $p_i(y)$, it holds that $p_{Y|X}(y \mid \mathbf{x}) = p_i(y)$ for all $\mathbf{x}$ with $e_i(\mathbf{x}) = 1$ and $p_X(\mathbf{x}) > 0$, then the induced density $p_{\mathcal{M}}(y \mid \mathbf{x})$ equals the true conditional density $p_{Y|X}(y \mid \mathbf{x})$.*

We provide a proof of Proposition 2 in see Appx. B.2. This result implies a structural constraint on any valid set of explanations. To guarantee a set of explanations induces the true conditional density sufficiently well, the explanations must partition the feature space, such that within the scope of each explanation $e_i$, the target variable $Y$ is i.i.d.

While this is a stronger requirement than for the marginal XMM, where a partitioning alone is sufficient, it also serves to eliminate degenerate solutions. By maximizing the conditional likelihood, the XMM is encouraged to find a set of explanations that capture where distinct, yet locally homogeneous sub-distributions occur. Therefore, we propose to fit an XMM $\mathcal{M}$ by minimizing the negative log-likelihood (NLL) given a dataset $\{(\mathbf{x}^{(l)}, y^{(l)})\}_{l=1}^n$

$$\text{NLL}(\mathcal{M}) = -\sum_{l=1}^n \log \left( \sum_{i=1}^k w_i(\mathbf{x}^{(l)}) \, p_i(y^{(l)}) \right) \ . \tag{5}$$

In practice, we estimate each $p_i$ from the subset $\{l : r_i(x^{(l)}) = 1\}$ with appropriate smoothing (e.g., KDE bandwidth selection or Dirichlet priors for discrete $Y$) and add a small $\varepsilon > 0$ inside the logarithm for numerical stability. This now provides a principled objective to learn an informative XMM using likelihood maximization. Once obtained, an XMM $\mathcal{M}$ gives insight into the global distribution $P_Y$ through its explainable components, and can also be used to make local, conditional density inferences $p_{\mathcal{M}}(y \mid \mathbf{x})$ for a given descriptive feature vector $\mathbf{x}$.

## 3.2 Optimization Objective

Lastly, we discuss how to optimize the NLL objective in Eq. 5 to learn an XMM. In Propositions 1 and 2, we have seen that an appropriate partitioning can achieve a perfect fit of the true density. On

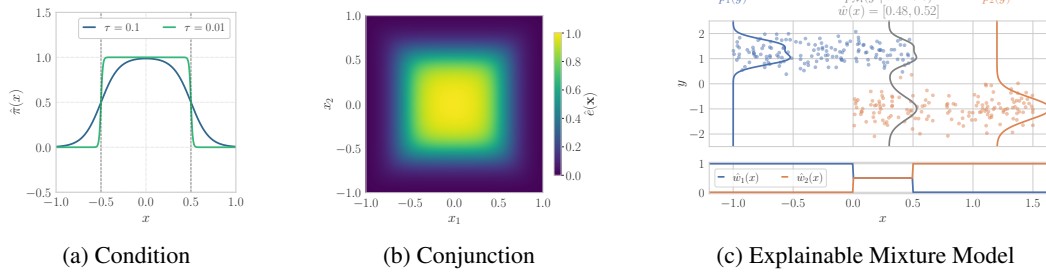

(a) Condition     (b) Conjunction     (c) Explainable Mixture Model

Figure 2: The building blocks of an XMM: Learnable thresholding conditions are placed on each feature $x_j \in \mathbb{R}$ (**a**). They are combined into a conjunctive, differentiable rule (**b**). Each rule acts as a gating function for an expert density, with a mixture in the overlap (**c**).

the other hand, we also want to allow a certain degree of overlap between explanations to improve interpretability, e.g. by providing broader, more general explanations.

To balance these two objectives, we propose to learn an XMM by minimizing a regularized NLL objective. We introduce an **overlap penalty** $\mathcal{R}(\mathcal{M})$ that penalizes explanations $e_i$ that frequently hold together. It is defined as

$$\mathcal{R}(\mathcal{M}) = \frac{1}{n} \sum_{l=1}^{n} \left( 1 - \sum_{i=1}^{k} w_i(\mathbf{x}^{(l)})^2 \right) . \tag{6}$$

For a particular sample $\mathbf{x}^{(l)}$, the term in parentheses is minimized when exactly one explanation $e_i$ holds, i.e. $w_i(\mathbf{x}^{(l)}) = 1$ for some $i$ and $w_j(\mathbf{x}^{(l)}) = 0$ for all $j \neq i$. To penalize overlap, we square the weights $w_i$ because the sum $\sum_{i=1}^{k} w_i(\mathbf{x}^{(l)}) = 1$ is constant by definition. Squaring ensures the penalty gets smaller as the distribution of weights becomes more sparse, and minimized when converging to a single active component. The overall optimization objective with a hyperparameter $\lambda$ that controls the strength of the overlap penalty is given by

$$\min_{\mathcal{M}} \text{NLL}(\mathcal{M}) + \lambda \mathcal{R}(\mathcal{M}) . \tag{7}$$

## 4    LEARNING EXPLAINABLE MIXTURE MODELS

In this section, we describe a concrete instantiation of Explainable Mixture Models for tabular data, which uses conjunctive rules as class of explanations, e.g. "18 < `Age` < 65 AND `BMI` > 25". Such rules, familiar from decision trees and subgroup discovery, offer a compact and human-interpretable description of the regions in which mixture components apply. Unlike combinatorial rule learning approaches, our method requires no pre-discretization; instead, the thresholds $\alpha_j, \beta_j$ are learned directly via gradient descent (see Eq. 9) for both continuous and discrete features. In particular, we consider rules $e : \mathbb{R}^d \to \{0, 1\}$ that map input features $\mathbf{x} \in \mathbb{R}^d$ to binary activations as per

$$e(\mathbf{x}; \theta) = \bigwedge_{j=1}^{d} \pi(x_j; \alpha_j, \beta_j) . \tag{8}$$

Learning a set of such rules is a challenging combinatorial problem, as the number of possible predicates, thresholds, and their conjunctions grows exponentially with the number of features $d$. To handle this complexity, we introduce a smooth relaxation that enables multiple rules to be learned jointly via gradient-based optimization.

### 4.1    A DIFFERENTIABLE RULE-BASED MIXTURE

We now show how to learn a rule-based mixture using gradient-based optimization. To avoid combinatorial search over an exponential search space (Lavrač et al., 2004; Atzmueller & Puppe, 2006), we extend the differentiable formulation of a single rule by Xu et al. (2024) to learn a mixture of multiple rules jointly using gradient-based optimization.

We briefly summarize the key components of the differentiable rule learner's architecture. Firstly, the conditions $\pi(x_j; \alpha_j, \beta_j) = \mathbb{1}[\alpha_j < x_j < \beta_j]$ placed on individual features $x_j \in \mathbb{R}, j \in \{1, \ldots, d\}$, are approximated as

$$\hat{\pi}_\tau(x_j; \alpha_j, \beta_j) = \sigma\left(\frac{x_j - \alpha_j}{\tau}\right) \sigma\left(\frac{\beta_j - x_j}{\tau}\right) , \qquad (9)$$

where $\sigma$ is the sigmoid function and $\tau > 0$ is a temperature parameter that controls its steepness. During training, we anneal the temperature gradually to zero, transitioning from soft constraints $\hat{\pi}: \mathbb{R} \to [0, 1]$ to hard constraints, i.e. $\lim_{\tau \to 0} \hat{\pi}_\tau(x_j; \alpha_j, \beta_j) = \pi(x_j; \alpha_j, \beta_j)$ for all $x_j \neq \alpha_j, \beta_j$. We show an example in Fig. 2a, where the condition becomes steeper as $\tau \to 0$.

We use the weighted harmonic mean to combine multiple conditions into a rule that approximates the logical AND operator. It is defined as

$$\hat{e}(\mathbf{x}; \theta) = \frac{\sum_{j=1}^d a_j}{\sum_{j=1}^d a_j \cdot \hat{\pi}_\tau(x_j; \alpha_j, \beta_j, \tau)^{-1}} \quad \text{with} \quad a_j \geq 0 , \qquad (10)$$

where we denote the parameters of a rule as $\theta = \{\alpha_j, \beta_j, a_j\}_{j=1}^d$. This function mimics the behavior of a logical conjunction whilst being fully differentiable: If any condition $\hat{\pi}_j(x_j)$ is close to zero, then the reciprocal $\hat{\pi}_j(x_j)^{-1}$ grows, and thus the overall rule activation $\hat{e}(\mathbf{x})$ becomes small. Conversely, the rule activation $\hat{e}(\mathbf{x}) = 1$ only if all conditions $\hat{\pi}_j(x_j) = 1$ are high. The learnable, non-negative weights $a_j$ represent the importance of feature $j$ within the rule. By setting $a_j = 0$, the corresponding condition $\hat{\pi}_j$ has no effect on the rule activation $\hat{e}(\mathbf{x})$, allowing the optimizer to effectively prune unnecessary conditions.

Last, we combine multiple differentiable rules with their local densities. To obtain an XMM, following Definition 2, we use as conditional gating function as follows

$$\hat{w}_i(\mathbf{x}; \Theta) = \frac{\hat{e}_i(\mathbf{x}; \theta_i) + \epsilon}{\sum_{j=1}^k \hat{e}_j(\mathbf{x}; \theta_j) + \epsilon} \quad \text{with} \quad \Theta = (\theta_1, \ldots, \theta_k) , \qquad (11)$$

for a given input $\mathbf{x}$, where we add an $\epsilon$ floor to avoid numerical instability. This formulation ensures that the mixture weights $\hat{w}_i(\mathbf{x}; \Theta)$ are non-negative and sum to one.

**Density Estimation.** To estimate the target density $p_i(y)$ for each component $i$, we need a density estimator $\hat{p}_i(y; \psi_i)$. We outline a parametric and a non-parametric solution that is then evaluated in the experiments. As a parametric alternative, we use an unconditional Gaussian mixture model (GMM). As we learn sub-distributions of the marginal distribution, we parameterize each component density $p_i(y)$ with the same set of means and covariances learned on the marginal distribution, but allow for different mixture weights $\psi_i$ for each component $i$. This has the advantage of being much more computationally efficient, and aligns with our goal of describing distinct modes in the data.

As the non-parametric variant, we use a Neural Spline Flow (NSF) (Durkan et al., 2019). A normalizing flow transforms a simple base distribution into a complex target distribution through a series of invertible mappings. NSFs are parameterized by a cubic spline neural network, whose parameters $\psi_i$ are learned by maximizing the likelihood. NSFs are powerful density estimators, but are computationally expensive and may overfit on small subgroups.

### 4.2 Overspecification and Pruning

A general challenge in rule learning is navigating the combinatorial search space of all possible rules. Previous approaches are limited to recursive partitioning or greedy schemes, but our differentiable approach allows for parallelized optimization of large quantities of rules. That is, we overspecify the initial number of rules $k$ to ensure sufficient coverage of the feature space.

To ensure that the initial rules effectively cover the feature space, the initialization of each rule is key. Random initialization of the rule parameters $\theta_i$ often leads to poor coverage (Fig. 3a), while choosing random samples from the training set as anchors improves coverage but can still leave gaps (Fig. 3b). We opt for a guided initialization, where we select as anchoring points k-means++ centroids (Arthur & Vassilvitskii, 2007) (Fig. 3c). This way, we ensure that each initial rule $\hat{e}_i$ is anchored on a distinct region of the feature space, improving the likelihood of discovering meaningful explanations.

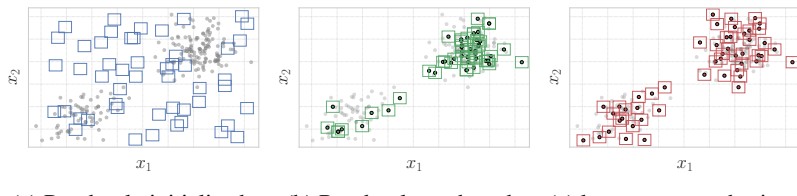

(a) Randomly initialized.    (b) Randomly anchored.    (c) k-means++ anchoring.

Figure 3: Initialization: k-means++ anchoring ensures a thorough coverage of the feature space.

**Pruning and Model Selection** Our initialization ensures broad coverage of the feature space, but overspecification inevitably introduces redundant explanations. The primary pruning mechanism is the optimization itself: a rule $\hat{e}_i$ can be disabled by learning an inverted interval ($\alpha_{ij} > \beta_{ij}$) for any feature $j$ with non-zero weight $a_{ij} > 0$, forcing $\hat{e}_i(\mathbf{x}) \approx 0$ everywhere and removing its gradient signal. This allows the optimizer to discard uncompetitive rules. For efficiency and stability, we periodically check for such inactive rules during training and disable them completely. If several neighboring rules converge to nearly identical densities $p_i(y)$, they may all survive pruning; we address this with a post-hoc merging procedure (see Appx. C.1).

While initializing with more components can reveal more specialized explanations, the gain in likelihood often comes at the cost of interpretability. To avoid dataset-specific tuning of the initial number of rules $k$, we use the Bayesian Information Criterion (BIC) to balance expressiveness and complexity. After training, we compute

$$\text{BIC}(\mathcal{M}) = 2 \cdot \text{NLL}(\mathcal{M}) + |\Theta| \log(n), \tag{12}$$

where $|\Theta|$ is the number of active parameters in the rule network. This criterion ignores parameters of the local density estimators $\hat{p}_i(y; \psi_i)$, as our framework models them to be data-induced, instead focusing model selection on the complexity of the explanations. We train multiple models from a range of $k$ and select the one with the best BIC score (see Appx. C.3).

## 5 EXPERIMENTS

We empirically validate XMM, using NSF and GMM respectively as density estimators. As baselines we include the interpretable CDE methods CDTREE (Yang & van Leeuwen, 2024) and CADET (Cousins & Riondato, 2019), which partition the feature space via decision trees, and non-interpretable methods MDN (Bishop, 1994), KMN (Ambrogioni et al., 2017), NF (Rezende & Mohamed, 2015), CVAE (Sohn et al., 2015) and LSCDE (Sugiyama et al., 2010). We provide the code for XMM, experiments and data generators in the Supplementary Material [1].

### 5.1 SYNTHETIC DATA

We first test on synthetic data with known ground truth. We generate $d$ independent uniformly distributed features $X_j$, partition the space into $k$ disjoint hyperrectangles, and assign each region a randomized density (Gaussian, Uniform, etc), resulting in a piecewise-constant $p(y \mid \mathbf{x})$ (see Appx.

---

[1] https://eda.group/prj/xmm/

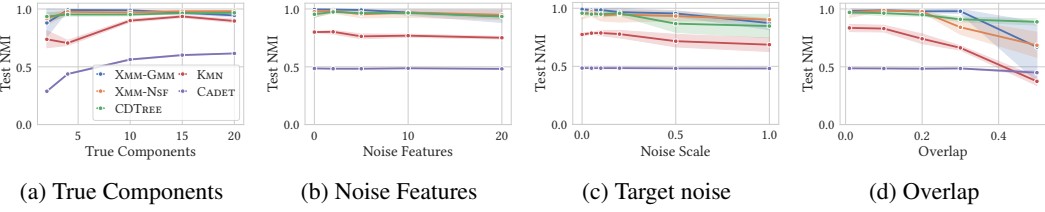

(a) True Components    (b) Noise Features    (c) Target noise    (d) Overlap

Figure 4: NMI between true and learned components across a variety of settings.

D.1). Unless varied as the experiment's parameter we use $d = 5$, $k = 5$ components, 600 samples per component, overlap $\beta = 0.1$ and no noise on $Y$, averaging results over 4 datasets.

**Accuracy.** We first measure the accuracy of XMM in recovering the ground-truth. We report the normalized mutual information (NMI), which compares the cluster similarity between true component labels and those by learned rules (Appx. D.4). Fig. 4a shows that both XMM instantiations reliably recover ground-truth components, with only slight performance drop for many components. CADET struggles due to unregularized large trees, while CDTREE regularization aids it in recovering a good solution. KMN, a black-box method from which we extract labels as that of the component with highest weighted likelihood, performs well on large numbers of components, but poorly on few.

**Robustness.** Second, we evaluate robustness to noise in the features and target, shown respectively Figures 4b and 4c. XMM is largely unaffected by feature noise and only slightly degrades under high target noise. CDTREE performs similarly but is less accurate at high target noise, while CADET and KMN are consistently weaker in both settings. In addition, we measure the effect of increasing overlap between the component densities in Fig. 4d. XMM remains stable under moderate overlap but degrades when overlap is large. KMN shows a similar trend, whereas CDTREE declines more gracefully and surpasses XMM at high overlap. CDTREE's advantage is its tendency to create many small leaves, which approximate overlapping densities well but are not penalized by the NMI metric.

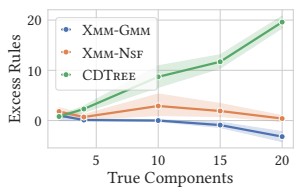

Figure 5: Excess rules vs. true components.

**Model Complexity.** Next, we assess model complexity by comparing the number of learned components to the ground truth. In Figure 5 we plot the number of excess components, i.e., the difference between learned and true components. Fig. 5 shows that after pruning, both XMM variants recover component counts close to the ground truth, with GMM slightly underfitting and NSF slightly overfitting. In contrast, the gap between CDTREE and the true number of components widens as complexity increases, reflecting the limitations of greedy top-down splitting, while CADET's number of excessive rules consistently exceeds the limits of the plot. On the other hand, XMM precisely identifies the correct number of components no matter if we have 5, 10, or 20 true components.

**Sensitivity to $\lambda$.** We further investigate the effect of the overlap penalty weight $\lambda$ (Eq. 7) on model complexity using the real datasets (Section 5.2). Figure 6a shows the change in test NLL ($NLL_\lambda - NLL_{\lambda=0}$) and Figure 6b shows the ratio of active rules relative to the unregularized baseline ($\lambda = 0$) As shown in Fig. 6, increasing $\lambda$ effectively regularizes XMM-GMM, using up to 16% fewer rules at $\lambda = 0.3$ than the baseline. The likelihood cost is negligible, indicating the components were redundant. This confirms that the penalty successfully steers the

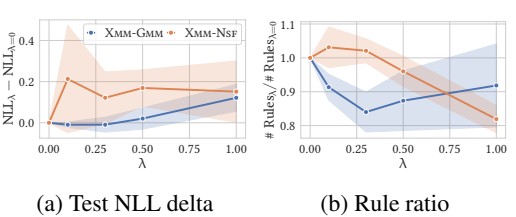

(a) Test NLL delta        (b) Rule ratio

Figure 6: Sensitivity to $\lambda$

optimization towards a concise partitioning for XMM-GMM. For XMM-NSF the benefit is less clear. The number of rules only decreases significantly at $\lambda = 1$ and incurs a larger likelihood cost. Consequently, we recommend the use of the overlap penalty primarily for the XMM-GMM variant.

**Rule Scaling.** Finally, we analyze the robustness to overspecification by increasing the initialized rules $k$ on synthetic datasets with 5 and 10 true components, and show the results in Figure 7. On these datasets we see in Figure 7a that once k is sufficiently large to capture the true structure, NLL plateaus. In Figure 7b and Figure 7c we see that XMM-GMM is very stable in this setting even when k is much larger than the true components, as no excess rules are discovered and NMI remains high. XMM-NSF achieves better NLL because it is more flexible, but this flexibility makes it more prone to retain excess rules when k is large. This indicates that the inductive bias of a restricted model class (XMM-GMM) allows for more effective pruning of excess rules through our likelihood objective.

| | Interpretable | | | | | | Black Box | | | | |
|---|---|---|---|---|---|---|---|---|---|---|---|
| Dataset | Xmm-Nsf | Xmm-Nsf Bic | Xmm-Gmm | Xmm-Gmm Bic | CDtree | Cadet | CVAE | Kmn | LsCde | Mdn | Nf |
| SkillCraft | −3.58 | −3.36 | −4.11 | **−4.19** | −4.03 | 2.23 | 1.61 | −0.94 | 1.57 | 2.73 | 1.47 |
| Thermography | 1.21 | 1.26 | 1.00 | 0.63 | **0.56** | 1.50 | 0.61 | 1.63 | 0.90 | 0.57 | 1.33 |
| abalone | −1.06 | −0.97 | **−2.73** | −2.72 | −2.20 | 4.32 | 1.92 | 1.89 | 2.13 | 1.88 | 1.79 |
| air quality | 0.27 | 0.27 | **−0.19** | **−0.19** | 0.53 | 1.40 | 0.15 | 0.25 | 0.91 | 0.18 | 0.15 |
| bike | 8.81 | 8.90 | 8.96 | 8.94 | 8.66 | 9.30 | 8.62 | 9.49 | 8.67 | **8.39** | 9.74 |
| boston | 2.98 | 2.99 | 2.60 | **2.58** | 2.93 | 5.51 | 3.20 | 3.17 | 3.07 | 2.67 | 7.32 |
| concrete | 3.64 | 3.61 | 3.50 | 3.73 | 3.58 | 3.54 | 3.11 | 3.33 | 3.61 | **2.96** | 3.45 |
| energy | 2.85 | 2.84 | 3.02 | 3.02 | 2.91 | 3.02 | 2.84 | 2.84 | 3.37 | 2.79 | **2.72** |
| insurance | 8.83 | 8.95 | 9.06 | 9.06 | 9.11 | 20.66 | 8.03 | 8.72 | 9.93 | 8.03 | **7.44** |
| life | 2.40 | 2.35 | 2.28 | 2.42 | 2.48 | 4.24 | 2.27 | 2.18 | 2.65 | **1.91** | 3.74 |
| obesity | −3.66 | −3.43 | **−4.86** | −4.53 | −3.45 | - | −0.18 | −1.78 | 1.12 | 2.76 | −0.39 |
| synchronous | −2.23 | −2.16 | −2.03 | −1.88 | −2.33 | −2.90 | **−4.80** | −2.41 | −1.25 | −3.08 | −4.11 |
| toxicity | 1.57 | 1.62 | 1.44 | 1.44 | 1.54 | 1.71 | **1.34** | 1.90 | 1.37 | 1.44 | 1.55 |
| wages | 10.88 | 11.13 | 10.89 | **10.80** | 11.20 | 11.90 | 11.33 | 11.68 | 11.45 | 11.59 | 11.53 |
| wine | −4.15 | −2.61 | **−4.91** | −4.89 | −4.61 | - | 1.15 | −1.37 | 1.20 | 3.29 | −0.38 |
| Rank | 5.60 | 6.07 | **4.20** | 4.47 | 5.20 | 9.73 | 4.80 | 6.60 | 8.07 | 4.73 | 6.33 |

Table 1: NLL of interpretable and black-box models on real-world datasets. Bold values indicate the best NLL among interpretable models, underlined values indicate the best overall NLL.

## 5.2 REAL-WORLD DATASETS

We next evaluate XMM on real-world datasets from the UCI Machine Learning Repository (Dua & Graff, 2017). Since ground-truth components are unavailable, performance is measured by negative log-likelihood (NLL) on a held-out test set. We report results using the full $k = 100$ starting components, as well as with BIC regularization for automatic model selection (Section 4.2).

We report the NLL in Table 1. XMM-GMM ranks highest across both interpretable and non-interpretable baselines, while the BIC-regularized variant achieves the second best rank but with substantially fewer and simpler rules (see Table 2). Among tree-based methods, CDTREE outperforms CADET and falls between our GMM and NSF instantiations. Non-interpretable methods vary in performance, with MDN and CVAE the strongest, but still trailing XMM-GMM. Overall, XMM achieves state-of-the-art accuracy with full interpretability. The XMM-GMM consistently outperforms XMM-NSF, suggesting that the simpler parametric estimator is better suited for this setting. BIC regularization typically incurs a small loss in accuracy but yields models with fewer, shorter rules, offering a practical trade-off between accuracy and interpretability.

## 5.3 CASE STUDY

We conclude with a case study on data-driven materials science. In particular we consider gold nanoclusters, whose electronic and catalytic properties are relevant to photovoltaics and medicine (Goldsmith et al., 2017). We fit an XMM to understand which molecular configurations lead to desirable properties. We target the HOMO-LUMO energy gap, a key indicator of photovoltaics performance, and visualize the learned densities and explanations in Fig. 8. We recover the known relationship that clusters with an odd number of atoms exhibit smaller gaps than those with an even number of atoms, but more importantly uncovering finer distinctions based on planarity, cluster size, and bonding structure. Compared to CDTREE, which requires 64 components for a weaker fit, XMM achieves a lower NLL (−1.706 vs. −1.683), with far fewer explanations (19.7 vs. 58.7) and orders-of-magnitude lower runtime (29s vs. 1782s).

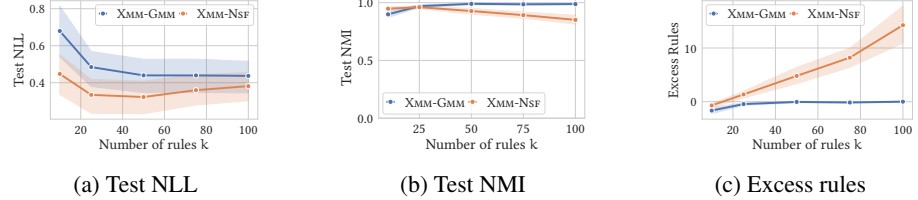

(a) Test NLL  (b) Test NMI  (c) Excess rules

Figure 7: Robustness to rule overspecification (large $k$). While XMM-NSF achieves lower NLL (a), it retains redundant rules (c). XMM-GMM successfully prunes excess components, maintaining high NMI (b) and recovering the exact number of ground-truth rules even as $k$ increases.

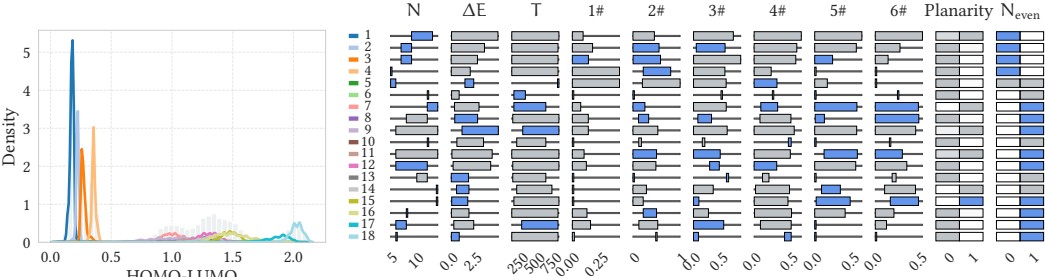

Figure 8: Densities and explanations for 18 mixture components learned by XMM. Continuous intervals are represented as bars relative to the feature domain, discrete values as boxes. Blue bars indicate active rule constraints ($a_j > 0$), gray ones indicate inactive features($a_j \leq 0$).

**Multi-Target Learning.** A distinctive feature of XMM is its capacity to explain multivariate targets. XMM identifies visible clusters in the joint distribution of relative gyration $R_{g0}$ and van der Waals energy $\Delta E_{vdW}$ in Fig. 9, revealing a clear separation in gyration $R_{g0}$ between planar (2D, Planarity $= 0$) and non-planar (3D, Planarity $= 1$) clusters. This matches the physical intuition that planar clusters are less compact and therefore have a larger radius of gyration. Our results further corroborate previous studies showing that non-planar clusters have higher intermolecular van der Waals interactions than planar ones (Goldsmith et al., 2017). For example, explanations 4 and 15 correspond to clusters of the same size but different planarity, yielding distinct $\Delta E_{vdW}$ values. Our results on real-world datasets, including a study on Abalone (see Appx. D.8), highlight the ability of XMM to explain meaningful interactions behind interesting subpopulations.

## 6 CONCLUSION

We introduced Explainable Mixture Models, a framework that pairs each mixture component with a human-interpretable rule. We established conditions for the exact recovery of the underlying data distribution, and proposed a scalable, differentiable learning algorithm with automatic model selection. Experiments show that XMM reliably recovers ground-truth components, while achieving state-of-the-art performance in CDE on real-world datasets. Case studies on materials science further illustrate the utility of XMM in AI for Science. Overall, XMM accurately models complex distributions whilst providing meaningful, interpretable explanations.

**Limitations.** A primary limitation of our approach is the need for a fixed number of mixture components $k$ at the start of training. We mitigate this through our initialization strategy and the BIC-based model selection, but in practice $k$ must be tuned for optimal results. Furthermore, we consider a limited class of explanations in the form of conjunctive rules over intervals. Future work will explore more expressive rule classes, such as disjunctive normal form rules, and extend explanations to different modalities such as images or text. Lastly, XMM is dependent on the performance of the underlying density estimator, which may need to be adapted to the specific data domain.

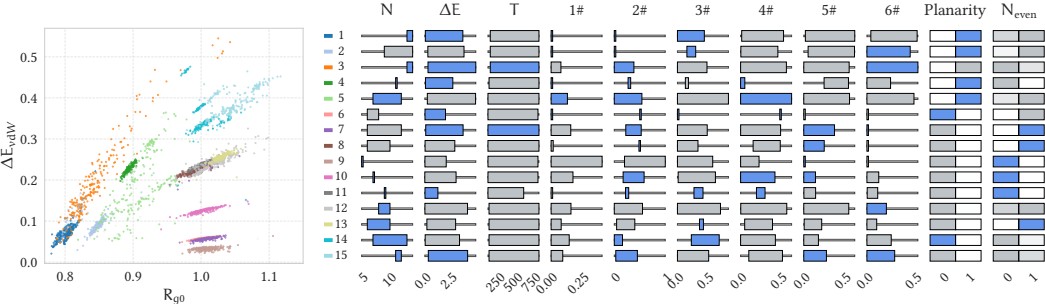

Figure 9: XMM over joint distribution of radius of gyration $R_{g0}$ and van der Waals energy $\Delta E_{vdW}$.

ETHICS STATEMENT

Our work aims to increase the transparency and interpretability of complex data distributions. The rules generated by our model are based on statistical correlations in the data and cannot be used to make definitive statements about causality or generalizability. The results must thus be used with caution, especially when sensitive data is involved.

REPRODUCIBILITY STATEMENT

To facilitate reproducibility we provide all code necessary to replicate the experiments. In addition to the method itself, this includes code to generate the synthetic data for our experiments, as well as code to reproduce the evaluation results on synthetic data, real data, and case studies.

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

APPENDIX

## A   GMM VS. XMM ON INSURANCE DATASET

On a widely used dataset of insurance charges (Choi, 2017) for example, a Gaussian mixture model (GMM) identifies subpopulations with distinct modes as shown in Fig. 10a. In many applications, however, we also have access to descriptive features, e.g. age or BMI. A fitted XMM in Fig. 10b recovers similar modes to the GMM, and additionally provides simple, interval-based rules that explain when each mode is observed. For example, the subpopulation with the lowest insurance charges corresponds to young, non-smoking individuals without children, whereas the highest-charge component comprises older, smoking individuals with high BMI.

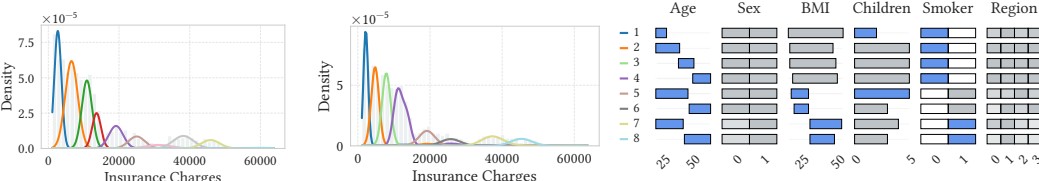

(a) The GMM recovers distinct modes, but no explanations.

(b) The XMM recovers similar modes to the GMM (left), and also explains when each mode is observed using simple rules over descriptive features.

Figure 10: Comparison of a Gaussian Mixture Model (GMM) and an Explainable Mixture Model (XMM) on a dataset of insurance claims (Choi, 2017).

## B   PROOFS

We provide the proofs for the propositions stated in the main text.

### B.1   PROOF OF RECOVERY OF MARGINAL DISTRIBUTION

**Proposition 1** *Let $\mathcal{M} = \{e_i\}_i^k$ be an XMM with a marginal density as per Def. 1. If the set of explanations $e_i$ form a partition of the feature space $\mathbb{R}^d$, i.e. $\sum_{i=1}^k e_i(\mathbf{x}) = 1$ for all $\mathbf{x}$ in the support of $P_X$, then the induced density $p_{\mathcal{M}}(y)$ equals the true marginal density $p_Y(y)$.*

**Proof:**   *By Definition 1, the induced marginal density of an XMM is*

$$p_{\mathcal{M}}(y) = \sum_{i=1}^k w_i\, p_i(y) \quad with \quad w_i = \frac{\mathbb{E}[e_i(X)]}{\sum_{j=1}^k \mathbb{E}[e_j(X)]}, \quad p_i(y) = p_{Y\,|\,(e_i(X)=1)}(y).$$

*If the explanations $\{e_i\}_{i=1}^k$ form a partition of the support of $P_X$, then $\sum_{i=1}^k e_i(x) = 1$ for all $x$ in the support of $P_X$, and hence*

$$\sum_{i=1}^k \mathbb{E}[e_i(X)] = \int_{\mathcal{X}} \sum_{i=1}^k e_i(x)\, p_X(x)\, dx = \int_{\mathcal{X}} p_X(x)\, dx = 1.$$

*Therefore $w_i = \mathbb{E}[e_i(X)]/1 = \mathbb{E}[e_i(X)]$, and substituting this yields*

$$p_{\mathcal{M}}(y) = \sum_{i=1}^k \mathbb{E}[e_i(X)]\, p_{Y\,|\,(e_i(X)=1)}(y)\,.$$

*By Bayes rule we rewrite*

$$p_{\mathcal{M}}(y) = \sum_{i=1}^k \mathbb{E}[e_i(X)]\, \frac{p_{Y,\, e_i(X)=1}(y)}{\mathbb{P}(e_i(X) = 1)} \tag{13}$$

As $\mathbb{E}[e_i(X)] = \mathbb{P}(e_i(X) = 1)$, *we can cancel terms to obtain*

$$p_{\mathcal{M}}(y) = \sum_{i=1}^{k} p_{Y, \, e_i(X)=1}(y) \, .$$

*Finally, since the events* $\{e_i(X) = 1\}_{i=1}^{k}$ *form a measurable partition of the support of* $X$, *the law of total probability implies*

$$\sum_{i=1}^{k} p_{Y, \, e_i(X)=1}(y) = p_Y(y).$$

*Thus* $p_{\mathcal{M}}(y) = p_Y(y)$, *proving the claim.*  $\square$

## B.2 PROOF OF RECOVERY OF CONDITIONAL DISTRIBUTION

**Proposition 2** *Let* $\mathcal{M} = \{e_i\}_i^{k}$ *be an* XMM *with a conditional density as per Def. 2. If the set of explanations* $e_i$ *form a partition of the feature space* $\mathbb{R}^d$ *into homogeneous regions with respect to the target variable* $Y$, *i.e. for every explanation* $e_i$ *and its induced sub-distribution* $p_i(y)$, *it holds that* $p_{Y|X}(y \mid \mathbf{x}) = p_i(y)$ *for all* $\mathbf{x}$ *with* $e_i(\mathbf{x}) = 1$ *and* $p_X(\mathbf{x}) > 0$, *then the induced density* $p_{\mathcal{M}}(y \mid \mathbf{x})$ *equals the true conditional density* $p_{Y|X}(y \mid \mathbf{x})$.

**Proof:**  *By Definition 2,*

$$p_{\mathcal{M}}(y \mid x) = \sum_{i=1}^{k} w_i(x) \, p_i(y), \quad w_i(x) = \frac{e_i(x)}{\sum_{j=1}^{k} e_j(x)}, \quad p_i(y) = p_{Y \mid (e_i(X)=1)}(y).$$

*If* $\{e_i\}_{i=1}^{k}$ *forms a partition of the feature space, then for every* $x$ *in the support of* $P_X$ *there exists a unique index* $i^{\star} = i^{\star}(x)$ *such that* $e_{i^{\star}}(x) = 1$ *and* $e_j(x) = 0$ *for all* $j \neq i^{\star}$. *Consequently,*

$$\sum_{j=1}^{k} e_j(x) = 1 \quad \Rightarrow \quad w_{i^{\star}}(x) = 1 \text{ and } w_j(x) = 0 \text{ for } j \neq i^{\star},$$

*and thus*

$$p_{\mathcal{M}}(y \mid x) = p_{i^{\star}}(y).$$

*By the homogeneity assumption of the proposition, for all* $x$ *with* $e_{i^{\star}}(x) = 1$ *we have*

$$p_{Y|X}(y \mid x) = p_{i^{\star}}(y).$$

*Combining the two displays yields* $p_{\mathcal{M}}(y \mid x) = p_{Y|X}(y \mid x)$ *for all such* $x$ *in the support of* $P_X$. *Hence the induced conditional density equals the true conditional density.*  $\square$

## C LEARNING AND OPTIMIZATION DETAILS

This appendix provides supplementary details on the training, optimization, and rule extraction procedures for XMM.

### C.1 ONLINE PRUNING AND POST-HOC MERGING

**Online Pruning.** During training, some rules may fail to specialize on any subset of the data. The optimizer can effectively disable such rules by learning an inverted interval ($\alpha_{ij} > \beta_{ij}$) for one or more of its predicates, which drives its activation $\hat{e}_i(\mathbf{x})$ towards zero. We periodically identify rules whose average mixture weight $\mathbb{E}_{\mathbf{x}}[w_i(\mathbf{x})]$ over the dataset falls below a small threshold (e.g., $10^{-3}$). These components are considered inactive and are permanently removed from the computation for the remainder of training by fixing $\hat{e}(\mathbf{x}) = 0$ and skipping density computation. This saves computational resources and improves stability by fully removing the gradient.

**Post-Hoc Merging.** The maximum likelihood objective is invariant to splitting a homogeneous data region into multiple sub-regions modeled by functionally identical experts. This can result in a fragmented solution. To improve interpretability, we merge such components after training. For all adjacent explanations $j, k$ we compute the pairwise similarity of the densities $\hat{p}_i(y)$ and $\hat{p}_j(y)$ using Jensen-Shannon divergence. We consider explanations adjacent if their data-based intervals (see Appendix C.2) touch ($\pm$ some tolerance) on one feature and are similar on all others with non-zero weight $a$. If the divergence between a pair of densities is below a predefined threshold, we merge their corresponding rules by taking the union of their data-based intervals and retain only one of the redundant experts.

## C.2 TEMPERATURE ANNEALING AND RULE EXTRACTION

To produce a final, human-readable set of rules, the soft, differentiable model must be converted into a discrete, logical representation.

**Temperature Annealing.** The temperature parameter $\tau$ in the soft predicate (Eq. 9) controls the trade-off between smooth gradients for effective optimization and sharp boundaries for interpretability. We begin training with a higher temperature to allow for a broader exploration of the solution space. As training progresses, we gradually anneal $\tau$ towards a small positive value. This process encourages the model to converge towards a solution with crisp, well-defined decision boundaries that closely approximate hard logical rules.

**Data-Based Rule Extraction.** Simply reporting the learned interval parameters $[\alpha_{ij}, \beta_{ij}]$ can be misleading, as optimization may push boundaries towards infinity in uncontested regions of the feature space. We therefore derive a more faithful representation of the learned partition from the empirical properties of the data governed by each rule.

For each explanation $e_i$, we first identify its corresponding data partition, $\mathcal{D}_i$. This partition consists of all samples assigned to component $i$ based on the maximum responsibility criterion, as defined for label extraction in Section D.4. That is,

$$\mathcal{D}_i = \{(\mathbf{x}, y) \mid i = \arg\max_j w_j(\mathbf{x})\} \ . \tag{14}$$

The final, human-readable rule for component $i$ is then defined by the empirical range of the data in $\mathcal{D}_i$ for each feature $j$: $[\min_{\mathbf{x} \in \mathcal{D}_i} x_j, \max_{\mathbf{x} \in \mathcal{D}_i} x_j]$. This data-derived bounding box is a valid representation because our predicate design ensures that if explanation $i$ has maximum responsibility for the empirical minimum and maximum values in $\mathcal{D}_i$, it also does so for all values in between. We report these ranges for all features, visually distinguishing those the model deemed unimportant (i.e., $a_{ij} \leq 0$) to communicate both the model's concise logic and the data's full distributional properties.

We use this this rule extraction to create the rule visualizations (see for example Fig 8). The bars indicate the range, categorical features show segments. The segments can be partially colored if multiple values are present in an explanation. Features that are active ($a > 0$) are blue, others are grey. The empirical intervals are computed for all features, active or not.

## C.3 MODEL SELECTION

Since the true number of components $k$ is unknown, we treat it as a hyperparameter. We train a set of models with a range of values for $k$ (e.g., $k \in \{10, 100\}$) and select the best one using the Bayesian Information Criterion (BIC). The BIC score is calculated after the online pruning and post-hoc merging steps have been applied. The penalty term in the BIC score considers only the number of active parameters in the gating network (the rule bounds $\alpha_{ij}, \beta_{ij}$ and weights $a_{ij}$). This choice reflects our goal of finding the most parsimonious partitioning of the feature space, rather than penalizing the complexity of the expert density estimators, which could otherwise dominate the score. This automatic balancing of model complexity and fit provides an alternative to manually chosing $k$.

# D  EXPERIMENTS

All experiments are performed on an Intel i5-12400 and Nvidia RTX 3070. GPU acceleration was used for methods that support it, which is true of XMM.

## D.1  SYNTHETIC DATA GENERATION DETAILS

We generate synthetic data from a process that mirrors our model's core assumption that the data arise from a mixture of components, where each component corresponds to a distinct subregion of the feature space with an associated conditional density. We define a collection of disjoint, axis-aligned hyperrectangular regions $\{H_j\}_{j=1}^k$ that partition the feature-space $\mathbb{R}^d$. For each region $H_j$, we define an unconditional target density $p_j(y)$ on $\mathcal{Y}$. The resulting ground-truth conditional density is piecewise-constant over $\mathbb{R}^d$, taking the value $p_j(y)$ for any feature vector $\mathbf{x} \in H_j$.

**Recursive Binary Partitioning.**   The regions are constructed by recursively splitting an initial hyperrectangle in a manner analogous to a decision tree. This procedure ensures that the resulting set of regions forms a true partition and avoids creating excessively thin regions. We also generate empty leaves that will not get any samples to make the data more realistic.

1. **Initialization.** Start with the full domain as the root of a tree.
2. **Recursive Splitting.** Iteratively select a leaf node and split it along a randomly chosen feature dimension. A split is permitted only if the node's width along that dimension exceeds a minimum threshold. The tree grows until a target number of leaves is reached.
3. **Component Selection.** From the set of leaf nodes, we select exactly $k$ to serve as the active components, defining the regions $\{H_j\}_{j=1}^k$.

We show a full partitioning in Fig. 11a and one that contains $50\%$ empty leaves in Fig. 11b.

**Conditional Density Assignment and Sampling.**   Once the feature space is partitioned, we assign target densities and generate samples. For each active region $H_j$, we draw an unconditional density $p_j(y)$ from a randomized family of standard distributions (Gaussian, Exponential, Gamma, Uniform) to induce diverse shapes. We show an example of such densities in Fig. 11c. To generate the dataset, we specify a fixed number of samples $n_j$ for each region. For each of the $n_j$ samples in region $H_j$, we first sample the feature vector $\mathbf{x}$ uniformly from within the hyperrectangle defining $H_j$, and then sample the target value $y$ from its corresponding density, $y \sim p_j(y)$. The resulting ground-truth conditional density is

$$p(y \mid \mathbf{x}) = \sum_{j=1}^k I\{\mathbf{x} \in H_j\} \, p_j(y) \, .$$

Task difficulty can be tuned by controlling the overlap between the densities $\{p_j(y)\}$ via a parameter $\beta \in [0, 0.5]$. A small $\beta$ yields well-separated densities, while $\beta = 0.5$ implies that all densities share the same median.

## D.2  BASELINE DETAILS

For all baseline methods, we utilized the authors' publicly available implementations and followed their recommended parameter settings unless otherwise specified.

**CDTREE.**   A state-of-the-art interpretable model that greedily builds a decision tree with non-parametric histogram densities in the leaves, regularized by the Minimum Description Length (MDL) principle. We use the authors' original R implementation with default parameters.

**CADET.**   An intrinsically interpretable CDE method that fits a decision tree with parametric distributions in the leaves. We use the authors' implementation with BIC regularization. The method requires specifying the parametric family for leaf distributions. We use Gaussians, as other families led to numerical failures on our test data. We further add very small Gaussian noise (standard deviation 0.001) to the target feature as duplicate values cause the method to fail.

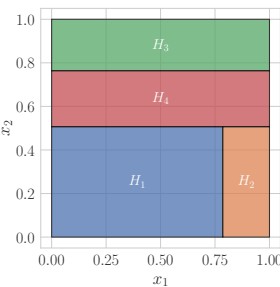 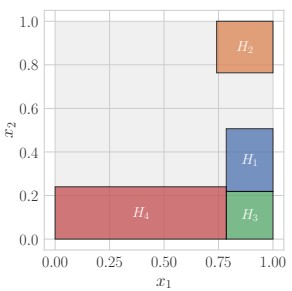 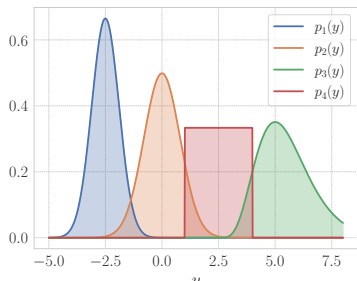

(a) Exact partitioning into 4 active regions.

(b) Partitioning with 4 active and 4 inactive regions.

(c) The constant densities $p_j(y)$ associated with each region $H_j$

Figure 11: Illustration of the steps involved in the synthetic data generation.

**Mixture Density Networks (MDN).**    A neural network-based approach where the network outputs the parameters (mixture weights, means, variances) of a Gaussian mixture model for the target variable, conditioned on the input features.

**Kernel Mixture Networks (KMN).**    Similar to MDN, but models $p(y \mid \mathbf{x})$ as a mixture of fixed kernel functions whose mixture weights are determined by a neural network conditioned on $\mathbf{x}$.

**Least-Squares CDE (LSCDE).**    A non-parametric method that directly models the conditional density without assuming a specific functional form, using a kernel-based approach.

**Normalizing Flows (NF).**    This method combines a conventional neural network with a multi-stage Normalizing Flow, where the neural network outputs the flow parameters.

For MDN, KMN, NF, and LSCDE, we use the implementations from the Python CDE package by Rothfuss et al.. We apply noise regularization of 0.01 to both features and targets, and otherwise use default parameters. On synthetic data we 3-fold cross validation to select the number of kernels of KMN to improve label quality.

**Conditional Variational Autoencoder (CVAE).**    We implement a CVAE (Sohn et al., 2015) with a learned conditional prior, where the encoder $q(z \mid \mathbf{x}, y)$, decoder $p(y \mid \mathbf{x}, z)$, and prior $p(z \mid \mathbf{x})$ are parameterized by multi-layer perceptrons with ReLU activations. We employ a latent dimension of 16, with hidden layer sizes of (128, 64, 32) for the encoder, (32, 64, 128) for the decoder, and (64, 32) for the prior. The decoder models the conditional likelihood as a Gaussian distribution. We optimize the Evidence Lower Bound (ELBO) with a KLD weight of 0.5 using Adam and apply early stopping based on validation set performance. Both features and targets are standardized during training. We estimate the test NLL by approximating the marginal likelihood $p(y \mid \mathbf{x})$ via Monte Carlo sampling with 2000 latent samples.

### D.3    IMPLEMENTATION AND PARAMETERS

We implement XMM in Python using standard machine learning libraries.

For the experiments we additionally apply a standard entropy loss regularizer to the feature importance weights. This mainly serves to make rules more concise for interpretability by encouraging the optimizer to actually reduce $a$ for redundant features. Let $\mathbf{a}_i = (a_{i1}, \ldots, a_{id})$ be the vector of non-negative feature importance weights for rule $\hat{e}_i$. Negative weights are set to 0 for this calculation. Rules with no support are ignored. First, these weights are normalized to form a probability distribution

$$\tilde{a}_{ij} = \frac{a_{ij}}{\sum_{l=1}^{d} a_{il}} \ . \tag{15}$$

| | Rule Complexity | | | | | | # Rules | | | | | |
|---|---|---|---|---|---|---|---|---|---|---|---|---|
| Dataset | XMM-NSF | XMM-NSF BIC | XMM-GMM | XMM-GMM BIC | CDTREE | CADET | XMM-NSF | XMM-NSF BIC | XMM-GMM | XMM-GMM BIC | CDTREE | CADET |
| SkillCraft | 10.17 | 9.83 | 7.07 | 6.67 | **0.00** | 6.66 | 6.00 | 6.00 | 15.00 | 6.00 | **1.00** | 166.00 |
| Thermography | 10.94 | 11.23 | 6.44 | 3.33 | **1.00** | 4.68 | 17.00 | 13.00 | 16.00 | **6.00** | 7.00 | 62.00 |
| abalone | 4.80 | 3.91 | 2.33 | 2.71 | **0.00** | 4.43 | 10.00 | 11.00 | 9.00 | 7.00 | **1.00** | 261.00 |
| air quality | 6.70 | 6.00 | 3.85 | 3.85 | **2.90** | 5.85 | 23.00 | 14.00 | **13.00** | **13.00** | 31.00 | 478.00 |
| bike | 8.53 | 8.31 | 5.19 | 3.73 | **2.83** | 4.02 | 17.00 | 16.00 | 31.00 | 11.00 | **6.00** | 45.00 |
| boston | 7.60 | 7.47 | 4.35 | 4.60 | **2.00** | 3.65 | 15.00 | 15.00 | 23.00 | 10.00 | **6.00** | 23.00 |
| concrete | 4.94 | 4.75 | 4.34 | 4.30 | **3.32** | 4.29 | 16.00 | 16.00 | 32.00 | **10.00** | 19.00 | 63.00 |
| energy | 2.67 | 2.67 | **1.88** | **1.88** | 2.56 | 3.77 | 33.00 | 21.00 | 8.00 | **8.00** | 34.00 | 598.00 |
| insurance | 4.19 | 2.91 | 2.91 | 2.91 | **2.69** | 4.38 | 16.00 | **11.00** | 11.00 | 11.00 | 13.00 | 85.00 |
| life | 9.73 | 10.00 | 7.33 | 7.12 | **2.90** | 4.60 | 11.00 | 12.00 | 18.00 | **8.00** | 20.00 | 102.00 |
| obesity | 9.00 | 7.86 | 5.17 | 3.86 | **0.00** | 4.23 | 7.00 | 7.00 | 23.00 | 7.00 | **1.00** | 127.00 |
| synchronous | 2.47 | 2.67 | 2.57 | 2.11 | **1.29** | 2.11 | 17.00 | 12.00 | 14.00 | **9.00** | 17.00 | 36.00 |
| toxicity | 4.27 | 4.29 | 3.84 | 3.43 | **1.67** | 3.43 | 15.00 | 14.00 | 25.00 | 7.00 | **6.00** | 53.00 |
| wages | 5.67 | 4.50 | 3.68 | 3.20 | **1.00** | 4.07 | 9.00 | 8.00 | 28.00 | 10.00 | **2.00** | 88.00 |
| wine | 5.00 | 5.00 | 4.67 | 3.75 | **0.00** | 6.55 | 5.00 | 3.00 | 9.00 | 4.00 | **1.00** | 300.00 |
| Rank | 5.33 | 4.87 | 3.27 | 2.33 | **1.13** | 3.53 | 3.47 | 2.60 | 3.80 | **1.73** | 2.47 | 5.93 |

Table 2: Rule complexity of interpretable models on real-world datasets.

The entropy regularization term is then the average Shannon entropy over all $k$ rules

$$\mathcal{R}_a(\mathcal{M}) = -\frac{1}{k} \sum_{i=1}^{k} \sum_{j=1}^{d} \tilde{a}_{ij} \log(\tilde{a}_{ij}) \tag{16}$$

Adding this to the objective we get

$$\min_{\mathcal{M}} \mathrm{NLL}(\mathcal{M}) + \lambda \mathcal{R}(\mathcal{M}) + \lambda_a \mathcal{R}_a(\mathcal{M}) \,, \tag{17}$$

where $\lambda_a$ is a hyperparameter.

For all experiments we use $\lambda = 0.1$. We use $\lambda_a = 0.05$ for synthetic experiments, and $\lambda_a = 0.1$ for the real data experiments. All synthetic experiments are ran with BIC selection of $k \in \{10, 100\}$, except the scaling experiment with $d = 20$ (Fig. 4a), where we use $k \in \{10, 200\}$. We always use a starting temperature of $\tau = 0.2$ and smoothly anneal it to $\tau = 0.005$ during the middle 80% of training epochs. The first and last 10% are reserved to encourage initial competition and final settling of the borders. For online pruning we use a threshold of $\mathbb{E}_{\mathbf{x}}[w_i(\mathbf{x})] \leq 0.005$.

### D.4 METRICS

**Component Label Extraction.** On synthetic data, we can compare the predicted component labels to the ground truth. For XMM, we assign each sample $(\mathbf{x}_n, y_n)$ to the component with the highest responsibility, which corresponds to the most active explanation for that sample's features:

$$\hat{z}_n = \underset{j \in \{1,\dots,k\}}{\arg\max} \, w_j(\mathbf{x}_n) \,. \tag{18}$$

For Kernel Mixture Networks (KMN), which models the conditional density as $p(y \mid \mathbf{x}) = \sum_{j=1}^{M} w_j(\mathbf{x}) \mathcal{K}(y; \mu_j, \sigma_j)$, we cannot obtain feature-based rules. Instead, we assign a label based on the most probable kernel component for the full data point:

$$\hat{z}_n = \underset{j \in \{1,\dots,M\}}{\arg\max} \, w_j(\mathbf{x}_n) \mathcal{K}(y_n; \mu_j, \sigma_j) \,. \tag{19}$$

### D.5 ADDITIONAL RESULTS

#### D.5.1 MODEL FIT ON SYNTHETIC DATA

**Pseudo $R^2$ ($R^2_{\mathbf{oracle}}$).** We report a normalized log-likelihood score to ensure comparability across different experimental settings. This metric measures the fraction of improvement a model achieves over an unconditional baseline, relative to the improvement achieved by the ground-truth data-generating model (oracle).

### D.6 DESTRUCTIVE NOISE

We perform an additional robustness experiment with destructive noise, showing the results in Figure 13. We replace the $Y$ value for an increasing fraction of samples with noise $\epsilon$ sampled from a

| Dataset | Interpretable | | | | | | Black Box | | | | |
|---|---|---|---|---|---|---|---|---|---|---|---|
| | Xmm-Nsf | Xmm-Nsf Bic | Xmm-Gmm | Xmm-Gmm Bic | CDTree | Cadet | CVAE | Kmn | LsCde | Mdn | Nf |
| SkillCraft | 305.9 | 255.4 | 29.5 | 39.7 | 3056.1 | **0.2** | 3.5 | 36.4 | 5.0 | 6.5 | 8.4 |
| Thermography | 678.3 | 517.5 | 32.2 | 46.0 | 217.4 | **0.1** | 6.2 | 32.8 | 2.3 | 5.3 | 5.5 |
| abalone | 398.5 | 335.3 | 22.3 | 35.1 | 9625.1 | **0.2** | 5.4 | 35.0 | 3.4 | 6.7 | 9.0 |
| air quality | 502.5 | 352.1 | 29.2 | 43.9 | 1385.8 | **0.5** | 17.6 | 39.6 | 5.7 | 9.0 | 11.7 |
| bike | 690.8 | 521.6 | 49.5 | 66.8 | 36.7 | **0.0** | 1.3 | 32.1 | 2.2 | 5.2 | 5.2 |
| boston | 610.6 | 485.0 | 43.9 | 54.6 | 59.9 | **0.0** | 0.7 | 31.4 | 1.1 | 4.5 | 5.2 |
| concrete | 772.3 | 524.5 | 51.3 | 59.6 | 95.2 | **0.0** | 3.3 | 32.3 | 2.0 | 5.0 | 5.5 |
| energy | 601.1 | 469.2 | 24.8 | 32.4 | 201.0 | **0.5** | 9.0 | 41.4 | 7.9 | 9.4 | 11.8 |
| insurance | 637.5 | 408.2 | 25.1 | 33.8 | 36.4 | **0.1** | 4.1 | 32.9 | 1.8 | 5.4 | 5.5 |
| life | 611.4 | 456.6 | 38.2 | 44.4 | 403.4 | **0.1** | 4.1 | 33.9 | 1.9 | 5.7 | 6.0 |
| obesity | 317.1 | 268.2 | 38.6 | 43.7 | 631.8 | **0.1** | 3.5 | 34.7 | 2.8 | 5.8 | 6.3 |
| synchronous | 608.4 | 456.6 | 27.3 | 35.6 | 45.8 | **0.0** | 5.2 | 31.9 | 1.4 | 4.4 | 5.2 |
| toxicity | 641.6 | 487.6 | 41.0 | 47.8 | 29.8 | **0.0** | 2.7 | 32.3 | 2.2 | 4.8 | 5.3 |
| wages | 588.8 | 419.0 | 42.1 | 50.8 | 62.8 | **0.1** | 2.5 | 33.5 | 2.4 | 5.4 | 5.6 |
| wine | 315.4 | 236.3 | 22.3 | 28.0 | 2168.6 | **0.3** | 6.7 | 36.1 | 5.3 | 6.9 | 9.9 |
| Rank | 10.7 | 9.7 | 6.6 | 8.0 | 9.3 | **1.0** | 3.1 | 6.7 | 2.2 | 3.8 | 4.9 |

Table 3: Runtime in seconds.

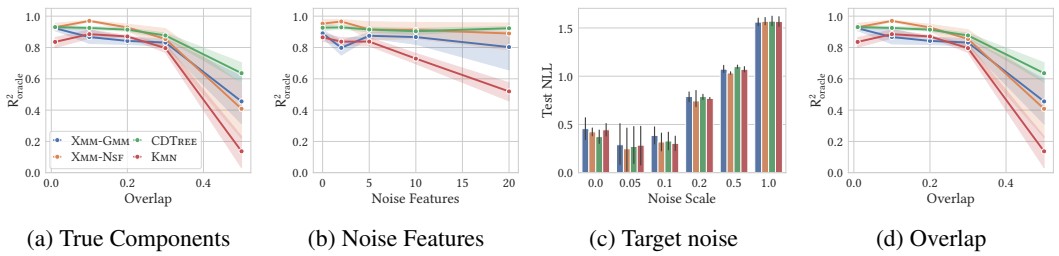

| (a) True Components | (b) Noise Features | (c) Target noise | (d) Overlap |
|---|---|---|---|

Figure 12: Likelihood fit ($R^2_{\text{oracle}}$) on synthetic data for varying (a) number of true components, (b) number of noise features, (c) target noise level, and (d) overlap between components.

Normal distribution $\epsilon \sim \mathcal{N}(\mu, 1)$ where $\mu = \mathbb{E}(Y)$. This tests robustness when noise introduces significant outliers relative to the true conditional distributions. In Figure 13b we see that the conditional structure is recovered accurately even when $30\%$ of $Y$ values are destroyed. Figure 13a shows the NLL. Due to the increased presence of outliers that are modeled by the same number of density estimators, the likelihood degrades when maintaining the true conditional structure.

### D.7 RUNTIME

Finally we evaluate the scalability as data dimensionality increases. For show the results for increasing $d$ in Fig. 14a and for increasing number of samples in Fig. 14b. We observe that neural methods like Xmm and KMN are consistently fast even on large datasets. Our NSF instantiation takes longer to run due to increased parameter count, but exhibits stable scaling. The runtime of CDTree increases very quickly even for moderate dimensions due to its iterative nature. CADET is comparatively very fast because of its small search space.

### D.8 ABALONE CASE STUDY

We apply Xmm to the popular abalone dataset which contains various size and weight measurments of abalones, a kind of sea snail. Typically this dataset is used for regression or classification using Age as the target variable. We apply Xmm using 28 Gaussian density components as there are 28 unique values in Age. In Fig. 15 we show that Xmm can recover reasonable explanations and distributions. The explanations show that larger and heavier abalones have a higher mean Age. But because we estimate the entire conditional distribution we can further see exactly how Age is distributed for these subgroups. For example explanations consisting mostly (1) or entirely (2) of infants are distributed in relatively low and narrow age range. Explanation 6 contains the largest and heaviest ones, which are distributed at the upper end with a wider distribution. We interpret this explanation to describe abalones that have reached their maximum size but continue to age. CDTree does not find any conditional structure in the data, returning a tree consisting only of the root node.

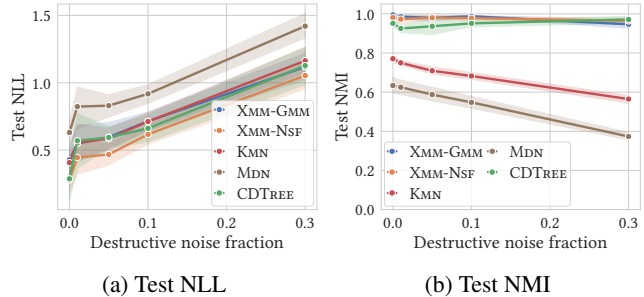

(a) Test NLL        (b) Test NMI

Figure 13: NLL and NMI for increasing fraction of $Y$ samples replaced with destructive noise.

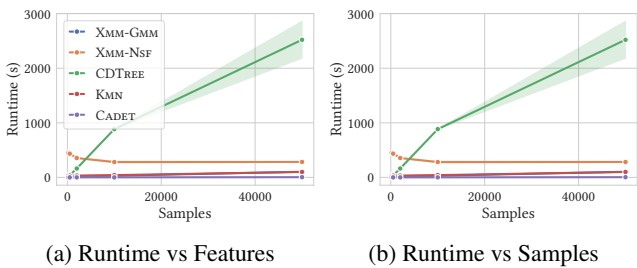

(a) Runtime vs Features        (b) Runtime vs Samples

Figure 14: Runtime of all methods on synthetic data with increasing number of features (left) and samples (right).

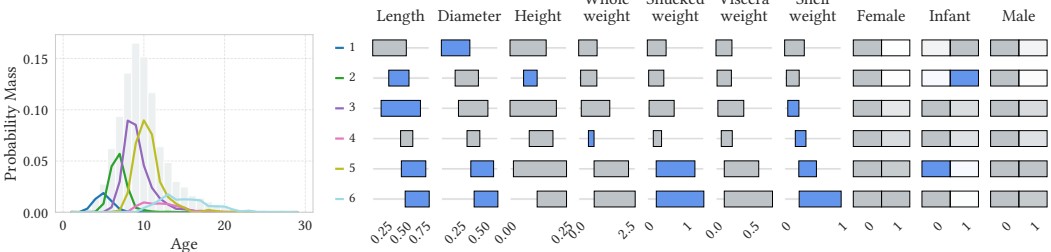

Figure 15: XMM results on Abalone. Probability masses are weighted by explanation size.

### D.9 GOLD HOMO-LUMO CDTREE

We provide a visualization of the CDTree density estimates on the Gold nano clusters dataset with target variable HOMO-LUMO in Figure 16.

## E LLM USAGE

LLM usage did not play a significant role in research ideation or writing of the paper itself. However LLMs and AI assistants were used during the implementation of the method.

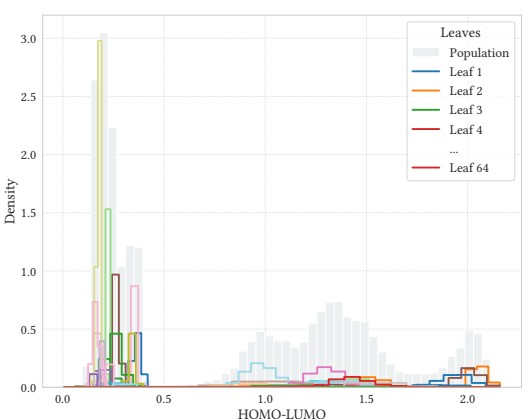

Figure 16: CDTree result on Gold dataset with HOMO-LUMO target. Densities are scaled by weight (relative number of samples per leaf). Legend abbreviated, colors repeat.

