# OpenReview forum: "Explainable Mixture Models through Differentiable Rule Learning"
_ICLR.cc/2026/Conference — ICLR 2026 Poster_

### Official Review · Reviewer_utZR · 2025-10-31

**Soundness:** 3
**Presentation:** 3
**Contribution:** 3
**Rating:** 6
**Confidence:** 2

**Summary:**

They propose a method for explainable mixture models. The explanations created are a conjunctions of different features. The mixture model created can fit the model well and able to generate logical rules to explain the mixtures created.

**Strengths:**

* Well written paper.
* Good exploration with synthetic dataset to showcase the capability of the models.
* Approach also fairly robust based on the synthetic dataset noise.

**Weaknesses:**

* The motivation of the problem with application can be improved.
* Many variants of their method is being introduced and it is a bit unclear on what the conclusion is between them. While they say that over NLL on real dataset is best for EMM-GMM, i am not sure how accurate it would be to claim that to be the best, as different datasets yielded different best model.
* Some human evaluation for some real-world dataset would have been good to evaluate the goodness of the explainability.

NOTE: i did not read the appendix section and my comments are purely based on main text.

**Questions:**

* Are the definitions in section 3 part of contribution or is it background?? It is a bit unclear. if it is background, then it is good to add ref. for them.
* What kind of noise is added in the robustness experiments and how much??
* In synthetic dataset will the performance change if the data is generated from a random distribution rather than a gaussian and uniform?
* Given the rules are logical statements can the features be continuous or do they have to be discrete/categorical for the approach to work?

---

> ### Author Response · Authors · 2025-11-20
>
> Dear Reviewer,
>
> Thank you for your constructive feedback. Below we answer your questions in detail.
>
> - **Definitions**: Definitions 1 and 2 are established in this paper and form the foundation of our proposed EMM framework. They are a central part of our contribution, and we will make this clear in the updated manuscript.
> - **Noise**:
> We consider two types of noise: We evaluate the robustness of EMMs against *additive* Gaussian noise in Figure 4c, that is noise on the target $Y$. We also test the robustness when adding independent, standard normal distributed noise features (Figure 4b). In both cases, we find EMMs are largely unaffected. We additionally report the results of an experiment with destructive noise below, where we replace a percentage of the points with randomly sampled points. We find that EMMs perform well under destructive noise in accurately estimating ground truth components, while as expected the data fit declines due to the noise. We will add this experiment to the updated manuscript.
>
>    **Normalized Mutual Information**
>    | Noise Fraction   |   0.0 |   0.01 |   0.05 |   0.1 |   0.3 |
>    |----------|-------|--------|--------|-------|-------|
>    | EMM GMM  |  0.99 |   0.99 |   0.98 |  0.99 |  0.95 |
>    | EMM NSF  |  0.98 |   0.97 |   0.98 |  0.98 |  0.96 |
>    | CDTree   |  0.95 |   0.93 |   0.94 |  0.95 |  0.97 |
>    | KMN      |  0.77 |   0.75 |   0.71 |  0.68 |  0.56 |
>
>    **Negative Log-Likelihood**
>    | Noise Fraction   |   0.0 |   0.01 |   0.05 |   0.1 |   0.3 |
>    |----------|-------|--------|--------|-------|-------|
>    | EMM GMM  |  0.43 |   0.55 |   0.6  |  0.71 |  1.12 |
>    | EMM NSF  |  0.28 |   0.45 |   0.47 |  0.62 |  1.05 |
>    | CDTree   |  0.29 |   0.57 |   0.59 |  0.66 |  1.13 |
>    | KMN      |  0.41 |   0.55 |   0.59 |  0.71 |  1.16 |
>    | MDN      |  0.63 |   0.82 |   0.83 |  0.92 |  1.42 |
>
> - **Data Generation**: In the synthetic experiments, the data is not limited to Gaussian or uniform distributions. Instead, for each component, we randomly sample from a pool of diverse distributions (Gaussian, Exponential, Gamma, Uniform), ensuring that the target mixtures cover a wide range of shapes and behaviors. Additionally, our approach is evaluated on multiple real-world datasets with naturally occurring, heterogeneous distributions, where it also performs well. This indicates that the method is robust across different distribution families.
> - **Features**: We allow both continuous and discrete input features, over which we construct each component's rule. Continuous features do not need to be discretized, as the thresholds are learned during optimization as per Eq. (9).

---

### Official Review · Reviewer_2m51 · 2025-11-01

**Soundness:** 3
**Presentation:** 3
**Contribution:** 2
**Rating:** 4
**Confidence:** 3

**Summary:**

This paper maximizes the conditional likelihood, where the conditional likelihood is parameterized using a mixture model, and adds a regularizer to the objective to encourage each data point to follow a unique path within the mixture. The authors design the mixture model as a rule function, providing explainability, and learnt by gradient descent.

**Strengths:**

The paper introduces a novel objective function that maximizes conditional density estimation in mixture-models.

**Weaknesses:**

How is this different from Mixture of Experts (MoE)?
MoEs use a gating function to decide which expert/block to activate, and are typically trained under a maximum likelihood objective for generative modeling. I see EMMs as a special case of MoEs where the gating logic is determined by learned rules. There are many possible gating mechanisms that can yield explainability, for example by restricting the gating function to be simple. Calling these models “EMMs” and not acknowledging them as a special case of MoEs is, in my view, misleading.

Definitions are not complete
How do you define “human-interpretable explanation”? This is subjective. You need to either define it explicitly or give application-specific criteria before introducing Definition 1 (Marginal-EMM). Otherwise, you cannot really call it a “definition.”

Also, from Definition 1, any tree-based method would qualify as an EMM if you take the path to the node as, $e_i$. Does that mean all tree-based models are EMMs?

Comparison to [1]
This paper modifies the objective proposed in [1] for conditional density estimation. Can you provide a direct comparison with [1]? The only clear difference I see is that your objective removes the entropy term. But [1] can still be used to compute a conditional density. How does the current method compare to [1] in terms of (a) explainability and (b) test-set likelihood?

I also disagree with the claim that tree-based approaches are “prone to overfitting.” Any machine learning method, deep learning models, are prone to overfitting.

Finally, in Eq. (6), since $w_i$ is constrained to be positive, why do you need to square it?

[1] Sascha Xu, Nils Philipp Walter, Janis Kalofolias, and Jilles Vreeken. “Learning Exceptional Subgroups by End-to-End Maximizing KL-Divergence.” ICML, 2024.

**Questions:**

Can you perform an ablation on $\lambda$, and also explain how increasing the number of mixtures affects the conditional likelihood?

---

> ### Author Response · Authors · 2025-11-20
>
> Dear Reviewer,
>
> Thank you for your constructive feedback. Below we address your main points in detail.
> - **Mixture of Experts**: We agree that EMMs share the same high-level functional form as MoEs: both are conditional mixtures $\sum_{i} \pi_i(x)p_i(y)$. However, EMMs differ in several fundamental and intentional ways.
>
>    Classical MoEs employ a general-purpose gating network (typically a neural network) and allow experts to take arbitrary forms. In contrast, an EMM is built from tuples of interpretable rules and their associated, data-derived local densities. Interpretability is not an optional constraint but a defining principle of the model. Existing interpretable MoE variants focus on predictive classification and do not provide differentiable rule learning or rule-based conditional densities [1,2].
>
>    EMMs therefore form a distinct subclass: they learn rules from an exponentially large combinatorial space, define components on human-understandable supports, and include pruning to maintain compactness and interpretability. These capabilities are not available in standard MoEs, which cannot recover such rule-structured solutions without imposing the constraints we introduce. For this reason, we refer to our model as an EMM rather than a generic MoE, and we will make this relationship and distinction explicit in the revised manuscript.
> - **Definition 1**: We agree that what amounts to a “human-interpretable” is inherently subjective and therefore intentionally avoid this term within the formal statement of Definition 1. The purpose of Definition 1 is to specify the mathematical framework of an EMM. To that end, we define explanations abstractly as indicator functions $e_i:\mathbb{R}^d\to \{0,1\}$, which determine whether a given input $x$ belongs to component $i$. This abstraction ensures that the definition itself is general, complete, and independent of any particular interpretability notion. The choice of which explanation class to use, e.g., conjunctive rules for tabular data, is an application-level design decision, such that practitioners are free to use explanation classes $e_i$ that are suited to their requirements and domains.
> - **Previous work**: The method by Xu et al. is an approach for subgroup discovery that finds a *single* rule that selects a part of the population for which the target variable exhibits an *exceptional* distribution. It does not provide a *global* density model. In contrast, EMMs decompose the entire feature space into multiple, explained regions and construct a complete mixture model for $p(y|x)$.
>
>    Methodologically, we adopt their differentiable formulation for *one* rule that selects *some* data, and generalize it to *multiple* explanations that together cover the *entire* data. To that end, we introduce a new architecture, complete with initialization, regularization, and sparsification methodology. Regarding the objective, Xu et al. maximize the KL divergence between the subgroup and the marginal. As such, their objective is not equivalent to maximizing the data likelihood as we do, and their model does not produce a test-time density estimator outside the discovered subgroups. We will clarify these differences in the paper.
> - **Tree-based methods**: Our remark refers specifically to the empirical behavior of CADET and CDTree in our experiments. Both methods showed signs of overfitting on the evaluated datasets. We agree that any ML method can overfit, and we will rephrase the statement to carry the intended message.
> - **Eq. 6**: The squared term is necessary to penalize overlapping activations. Because mixture weights fulfill $\sum w_i(x)=1$ by design (Definition 2), without a square, any mixture fulfills $\mathcal{R}(M)=0$. On the other hand, by squaring $w_i(x)^2$, a global minimum is achieved only with one $w_j(x)=1$ and all other $w_k(x)=0$, which is the intended behavior
>
> [1] Ismail, A., Arik, S. Ö., Yoon, J., Taly, A., Feizi, S., & Pfister, T. (2023). Interpretable Mixture of Experts. Transactions on Machine Learning Research, 12, 1–28.
> [2] Pradier, M. F., Zazo, J., Parbhoo, S., Perlis, R. H., Zazzi, M., & Doshi-Velez, F. (2021). Preferential mixture-of-experts: Interpretable models that rely on human expertise as much as possible. AMIA Summits on Translational Science Proceedings, 2021, 525.

---

> > ### Author Response · Authors · 2025-11-20
> >
> > We also conducted additional ablation studies to further strengthen the evaluation of EMM.
> > - **Ablation on $\lambda$**: We report the difference in likelihood and number of rules when disabling the regularization for the real-world datasets in the tables below. $\lambda$ values between [0.1,0.3] result in fewer rules (simpler model) at no NLL cost with the GMM variant,  indicating that the regularization successfully removes unnecessary rule components without impairing fit. For the neural spline flow (NSF) $\lambda=1$ is required to achieve simpler models, but incurs a penalty for the likelihood.
> >
> >    **Delta in negative log likelihood  to $\lambda=0$** (lower is better)
> >    |   $\lambda$  |   0.0 |   0.1 |   0.3 |   0.5 |   1.0 |
> >    |----------|-------|-------|-------|-------|-------|
> >    | EMM GMM  |     0 | -0.01 | -0.01 |  0.02 |  0.12 |
> >    | EMM NSF  |     0 |  0.21 |  0.12 |  0.17 |  0.15 |
> >
> >    **Ratio of found rules relative to $\lambda=0$** (lower is better)
> >    |  $\lambda$   |   0.0 |   0.1 |   0.3 |   0.5 |   1.0 |
> >    |----------|-------|-------|-------|-------|-------|
> >    | EMM GMM  |     1 |  0.91 |  0.84 |  0.87 |  0.92 |
> >    | EMM NSF  |     1 |  1.03 |  1.02 |  0.96 |  0.82 |
> > - **Number of Rules**: To analyze how the number of mixture components influences model performance, we varied the initial number of rules in synthetic experiments. Both data fit and component recovery improve as $k$ increases, up to around **50 components**, after which performance plateaus. This suggests that the model effectively utilizes additional capacity up to a point, beyond which further rules do not contribute meaningfully.
> >
> >    **Negative Log Likelihood**
> >    | $k$ initial rules  |   10 |   25 |   50 |   75 |   100 |
> >    |----------|------|------|------|------|-------|
> >    | EMM GMM  | 0.68 | 0.48 | 0.44 | 0.44 |  0.44 |
> >    | EMM NSF  | 0.45 | 0.33 | 0.32 | 0.36 |  0.38 |
> >
> >    **Normalized Mutual Information**
> >    | $k$ initial rules   |   10 |   25 |   50 |   75 |   100 |
> >    |----------|------|------|------|------|-------|
> >    | EMM GMM  | 0.9  | 0.97 | 0.99 | 0.99 |  0.99 |
> >    | EMM NSF  | 0.95 | 0.96 | 0.93 | 0.89 |  0.85 |

---

### Official Review · Reviewer_LLHS · 2025-11-01

**Soundness:** 2
**Presentation:** 3
**Contribution:** 2
**Rating:** 2
**Confidence:** 4

**Summary:**

This paper proposes EMM, explainable mixture model, a principled learning procedure that discovers each mixture component with a human-interpretable rule over descriptive features. Experiments on synthetic and real-world datasets demonstrate the effectiveness of the proposed approach.

**Strengths:**

The proposed approach is based on a principled machine learning procedure.

**Weaknesses:**

The experiments are restricted to mixture models, and don't reflect the paradigm shift in the era of deep learning. The experiments should include the results that compare with deep learning models such as VAE, and whether the proposed can be extended deep learning models; the data sets are UCI which are too small, don't support the claim that the proposed method is scalable and applicable to more complex, multi-model distributions.

**Questions:**

N/A

---

> ### Author Response · Authors · 2025-11-20
>
> Dear Reviewer,
>
> There seems to be a misunderstanding with regarding to VAEs and EMMs. We here clarify the difference. VAEs model the *marginal* distribution $p(x) = \int_z p(x\mid z) p(z)$ using a latent prior over $z$. In contrast, EMMs model the *conditional* density of the target variable $Y$ by learning explicit, rule-based components over a separate feature space $X$. That is, VAEs and EMMs focus on fundamentally different problems: (latent) representation learning vs interpretable mixture/conditional density learning.
>
> In our evaluation, we already included state-of-the-art deep learning based density estimators in the form of normalizing flows. Furthermore, we also compare to Kernel Density Networks and Mixture Density Networks. These are implemented using Multi-Layer Perceptrons and go well beyond simple mixture models. We agree that additionally comparing to a conditional VAE-based approach [1] makes sense. We provide the results on the real-world datasets in the table below. Overall, although the CVAE is able to fit more accurate models for certain datasets, it completely lacks the ability to return interpretable descriptions for its components – the main point of EMMs.
>
> For our qualitative comparisons, we focus on interpretable conditional density estimation methods, specifically the tree-based CDTree and CADET, because they are the only existing methods that can name and describe their components. Our method, EMMs, follows the broader paradigm shift towards continuous optimization by using a fully differentiable rule learning procedure, whilst still yielding a rule-based, human-interpretable representation.
>
> Lastly, we do not use just “UCI” datasets, but a variety of synthetic, real-world and domain-specific benchmarks that quantitatively and qualitatively highlight the advantages of EMMs over existing approaches. We benchmark synthetic datasets with up to $100$ features and up to 50,000 samples.  In particular, for the gold nanocluster datasets with a dimension of $24000 \times 12$, we demonstrate that EMMs can decompose a multivariate target variable into distinct, rule-based clusters, a capability that, to the best of our knowledge, was not previously possible.
>
> In summary, we hope this clarifies (i) the conceptual distinction between EMMs and VAEs, (ii) the breadth and appropriateness of the baselines used in our experiments, and (iii) the rationale for our focus on interpretable conditional density estimators in the qualitative analysis. EMMs offer a novel, rule-based decomposition of complex conditional distributions, achieving competitive predictive performance.
>
> [1] Sohn, Kihyuk, Honglak Lee, and Xinchen Yan. Learning structured output representation using deep conditional generative models. Advances in neural information processing systems 28 (2015).
>
>
>
> Test NLL
> | dataset          |   CDTree |   EMM GMM |   CVAE |   MDN |
> |------------------|----------|-----------|--------|-------|
> | SkillCraft.csv   |    -4.03 |     -4.11 |   1.61 |  2.73 |
> | Thermography.csv |     0.56 |      1    |   0.61 |  0.57 |
> | abalone.csv      |    -2.2  |     -2.73 |   1.92 |  1.88 |
> | air_quality.csv  |     0.53 |     -0.19 |   0.15 |  0.18 |
> | bike.csv         |     8.66 |      8.96 |   8.62 |  8.39 |
> | boston.csv       |     2.93 |      2.6  |   3.2  |  2.67 |
> | concrete.csv     |     3.58 |      3.5  |   3.11 |  2.96 |
> | energy.csv       |     2.91 |      3.02 |   2.84 |  2.79 |
> | insurance.csv    |     9.11 |      9.06 |   8.03 |  8.03 |
> | life.csv         |     2.48 |      2.28 |   2.27 |  1.91 |
> | obesity.csv      |    -3.45 |     -4.86 |  -0.18 |  2.76 |
> | synchronous.csv  |    -2.33 |     -2.03 |  -4.8  | -3.08 |
> | toxicity.csv     |     1.54 |      1.44 |   1.34 |  1.44 |
> | wages.csv        |    11.2  |     10.89 |  11.33 | 11.59 |
> | wine.csv         |    -4.61 |     -4.91 |   1.15 |  3.29 |

---

### Official Review · Reviewer_ZarE · 2025-11-04

**Soundness:** 3
**Presentation:** 2
**Contribution:** 3
**Rating:** 6
**Confidence:** 2

**Summary:**

Given a set of data-value pairs $(x_i,y_i)$, where the covariates $x_i \in \mathbb{R}^d$, the paper proposes an explainable mixture model (EMM) approximation of the conditional density $p(y|x)$. The mixture model is conceptually proposed as follows:
- first, the $p(y|x)$ is approximated by a weighted mixture of $k$ components
- each component ideally represented as a box (intersection of intervals) in $\mathbb{R}^d$ or in a lower dimension. The authors show that if the components are supported on a partition of $\mathbb{R}^d$, then there is no error in the approximation
- In order to apply standard optimisation to learn the parameters, each component is approximated by a  smoothed "box" (equation 10)
- the mixture model parameters are learned by minimising a regularised negative log likelihood
- some further pruning is done to reduce the number of rules learned

The authors demonstrate that the proposed EMM model provides good fit of data, while providing fewer learning rules for the explainable model.

The authors show

**Strengths:**

The overall EMM model uses a combination of simple ideas, but learns an explainable rule-based model that consistently has fewer rules.
The experiments look convincing (with the caveat that I not an expert in this specific problem)

**Weaknesses:**

The paper's theoretical contribution is limited. A simple result is provided in the case where the supports of the components form a partition. However, there are no guarantees regarding the severity of the approximation error in the general case.
I was tempted to give a low score due to this, but it turns out that this is not a concern in relevant literature (papers like CADET, CDTree).

I found parts of the papers a bit difficult to follow, which could be because I am not familiar with this line of work (and generally, less familiar with explainability literature).

**Questions:**

Could the authors explain how to read the explanation plots (for example, right hand side of Fig 7)? What do the bars and colours mean?

---

> ### Author Response · Authors · 2025-11-20
>
> Dear Reviewer,
>
> Thank you for your thoughtful assessment of our work.
> Your main point relates to the theoretical contribution, specifically the absence of approximation guarantees beyond the partition case. Our exactness result, however, already has important implications for what the optimal EMM solution will look like and, therefore what one can expect from the model: a decomposition of the conditional distribution into explainable, internally homogeneous components that together reconstruct the full density. Extending this guarantee to the general setting would require imposing strong smoothness or regularity assumptions on the true conditional density. These assumptions are typically unverifiable in practice and are not adopted in prior explainable density-estimation work (e.g., CADET, CDTree).
>
> Rather, our main contribution is the introduction of a principled framework for interpretable mixture modeling, which provides an explanation-driven decomposition into local, data-induced component distributions. Building on this foundation, we propose a fully differentiable learning method that can efficiently search an exponentially large space of candidate rules, select a compact combination thereof, and jointly optimize the associated densities. Despite this complexity, our method scales well and consistently discovers a small number of human-interpretable components across diverse datasets with strong likelihood performance but more compact than comparable approaches.
>
> The explanation plots in Figures 1b, 6, and 7 can be interpreted as follows: each feature $j$ may either be included in the rule (blue) or excluded (gray), which the method determines by learning to set $a_j < 0$. If a feature is included, the blue interval represents the condition imposed on that feature, i.e., $x_j \in [\alpha_j,\beta_j]$. For discrete features, we simplify the visual representation by marking the included values in blue. We will clarify this in the revised manuscript.

---

> > ### Comment · Reviewer_ZarE · 2025-11-25
> >
> > I thank the authors for their response. I am maintaining my marginally positive score because I appreciate the contribution, but it would be challenging for me to champion this paper.

---

### Author Response · Authors · 2025-12-01

In the revised manuscript, we have clarified several points raised by the reviewers and strengthened the experimental evaluation.
1. **Improved Experiments**: We expanded the experimental section to more thoroughly assess the empirical behavior of EMMs:
   - **Additional Baseline**: We now include a conditional variational autoencoder (CVAE) as a baseline. While CVAEs and EMMs achieve comparable likelihoods, the key distinction is that CVAEs yield latent components, whereas EMMs yield interpretable ones.
   - **Robustness to Destructive Noise**: We evaluate performance under varying levels of destructive noise and find that EMMs remain resilient across all tested noise intensities.
   - **Effect of Model Complexity**: We examine EMMs with increasing numbers of rules and observe that performance improves for an increasing number of rules until it plateaus at roughly 50 rules.
   - **Ablation Study on Regularization**: We provide an ablation of the regularization strength $\lambda$, showing that moderate values produce more compact EMMs with fewer components while maintaining identical goodness of fit to the unregularized model.
2. **Significance of Contribution**: EMMs target a key problem in exploratory data analysis: discovering from a heterogeneous population locally homogeneous demographic groups. Our framework formalizes this challenge as a mixture-learning task and establishes conditions under which EMMs exactly recover the true conditional distribution. Our result shows that likelihood maximization naturally leads EMMs to uncover meaningful, locally homogeneous subpopulations.

   Methodologically, we introduce a differentiable rule-ensemble architecture that learns more compact and more accurate conditional mixtures compared to existing CDE approaches. Across multiple real-world datasets from domains such as materials science, we demonstrate that EMMs yield precise, interpretable decompositions of complex data.
3. **Relationship to Mixtures-of-Experts (MoE)**:
Although EMMs and Mixtures of Experts are both mixture models with input-dependent weights, they differ in important ways. Standard Mixtures of Experts use flexible gating networks and allow experts to take any parametric form, with no requirement that these components correspond to meaningful or interpretable subgroups. Their primary aim is predictive performance, and nothing in their structure encourages the recovery of rule-based or human-understandable components. EMMs, in contrast, are built around explanation-based components defined by explicit rules together with their data-derived local densities. In our framework, interpretability is the central design principle rather than an optional feature.

---

### Meta-Review · Area_Chair_wr34 · 2026-01-06

**Summary:**

The reviewers all felt that the idea of learning interpretable rule-based mixture models is indeed interesting. The key concerns revolved around perceivably incremental contributions relative to mixture of experts, weak theoretical guarantees, older experiments and positioning. I had to fully skim the paper before I could provide an informed judgement. Before the rebuttal phase, this paper was clearly in the reject category since more work was needed. But in the rebuttal phase, the authors have clarified some important issues here.


The idea of mapping to logical rules is the key strength of this paper. And sparsity is another key advantage.

The theoretical guarantees are a bit weaker. I completely disagree about modern ML. I feel that the reviewer should be banned (more below). Better positioning w.r.t related work is important and some improvement in justification can greatly help the paper.

**Reviewer Concerns:**

The authors have clarified some parts of the paper. Specifically, I think the explanation w.r.t MoEs will have significantly improved the review scores. I think teh distinction between predictive performance vs learning interpretable rules is indeed interesting.

**Reviewer Scores:**

I think reviewer 2m51 would have increased the score a bit to a 5 at the very least.

I will completely drop reviewer LLHS from here. This is because of the standard, deep learning revolution and age and old ML does not work nonsense that they tend to write.

So given that it will be a 5 5 6, I think the paper deserves a poster.

---

### Decision · Program_Chairs · 2026-01-26

Accept (Poster)